# Alkaline-Based Catalysts for Glycerol Polymerization Reaction: A Review

**Negisa Ebadipour**, **Sébastien Paul**, **Benjamin Katryniok and Franck Dumeignil** *

Univ. Lille, CNRS, Centrale Lille, Univ. Artois, UMR 8181–UCCS–Unité de Catalyse et Chimie du Solide, F-59000 Lille, France; negisa.ebadi-pour@centralelille.fr (N.E.); sebastien.paul@centralelille.fr (S.P.); benjamin.katryniok@centralelille.fr (B.K.)

* Correspondence: franck.dumeignil@univ-lille.fr; Tel.: +33-(0)3-20-43-45-38

**Abstract:** Polyglycerols (PGs) are biocompatible and highly functional polyols with a wide range of applications, such as emulsifiers, stabilizers and antimicrobial agents, in many industries including cosmetics, food, plastic and biomedical. The demand increase for biobased PGs encourages researchers to develop new catalytic systems for glycerol polymerization. This review focuses on alkaline homogeneous and heterogeneous catalysts. The performances of the alkaline catalysts are compared in terms of conversion and selectivity, and their respective advantages and disadvantages are commented. While homogeneous catalysts exhibit a high catalytic activity, they cannot be recycled and reused, whereas solid catalysts can be partially recycled. The key issue for heterogenous catalytic systems, which is unsolved thus far, is linked to their instability due to partial dissolution in the reaction medium. Further, this paper also reviews the proposed mechanisms of glycerol polymerization over alkaline-based catalysts and discusses the various operating conditions with an impact on performance. More particularly, temperature and amount of catalyst are proven to have a significant influence on glycerol conversion and on its polymerization extent.

**Keywords:** glycerol polymerization; polyglycerol; alkaline catalyst; mechanism

## 1. Introduction

In the last few years, in view of the environmental issues faced by humanity, the use of biofuels has been favored to try to limit the emissions of greenhouse gases and hence the global warning effect. Within that frame, biodiesel production has been promoted with, as a consequence, a significant increase in glycerol availability, as it is coproduced with biodiesel with a ratio of ca. 100 kg of glycerol per ton of biodiesel. Even if the development of the biodiesel industry seems to be a bit slowed down due to recent political decisions in Europe, this sector still generates worldwide a huge amount of glycerol which must be valorized to help sustain the economy of the biodiesel value chain. Of course, the higher is the value of the chemicals derived from glycerol, the better.

In that context, polyglycerols (PGs) are very interesting candidates. Indeed, these polymers have demonstrated excellent biocompatibility. They currently have found direct applications in the pharmaceutical, cosmetic and food sectors. PGs are a good basis for obtaining other molecules such as PGs esters, which represent a global market size of USD 1.91 billion in 2017 [1]. Considering the large variety of final applications (emulsifiers, stabilizers, antimicrobial agent, cosmetics, etc.), a dramatic increase in the production is envisioned with a global market of USD 5.52 billion in 2022.

While polyglycerols are currently produced from biosourced epichlorohydrin (Epicerol®), the change in the customers' demand, more particularly in the cosmetic and food sectors, require products derived from nontoxic and preferably natural ingredients. This strongly encourages the manufacturers to produce PGs directly from glycerol. Currently, in Europe, manufacturers such as

Solvay Chemicals, Lonza and, recently, Spiga Nord produce PGs from refined glycerol deriving from the biodiesel and oleochemicals industries [2].

Here, it should be noted that in the literature glycerol polymerization is referred to as etherification [3–5], oligomerization [6,7] and polymerization [8]. In this review, we use the general term "polymerization" to refer to all these denominations.

The aim of this paper is thus to review recent advances in the field of glycerol polymerization, with a specific focus on alkaline-catalyzed reaction, which fills the gap with previous reviews where focused on heterogeneous and homogeneous catalysts and PG2 applications [9], as well as clay-based catalysts and their roles in the shape selectivity [10] with the recent advances made in this field. This paper also provides a view on polyglycerols applications, on the reaction mechanism involved in alkaline-catalyzed PGs reaction, with a focus on the issue of leaching and on the stability of these heterogeneous catalysts, as this critical point was not addressed by the previous reviews. Finally, the influence of the reaction parameters on catalytic performances is also addressed.

## 2. Polyglycerols Applications

Generally, polyglycerols are highly water soluble, biocompatible and highly functional materials, which, as said above, make them very interesting in many applications related to cosmetics, food, polymers and plastic industries [9,11–13]. In the current market, di-, tri-, tetra-, penta-, hexa- and decaglycerols (PG10) are available as "linear" PGs [9,11]. However, the applications of PGs depend on their degree of polymerization, number of hydroxyl group, structure and properties such as viscosity or functionality. Hydroxyl functionality refers to the number of primary OH groups and secondary OH groups per molecule that make them reactive for further modifications.

As shown in Figure 1, polyglycerols have a large variety of possible chemical architectures: linear, branched, cyclic and hyperbranched (identified by $^{13}$C NMR analysis). For instance, 12, 360 and 19,958,400 isomers could be formed for PG3, PG5 and PG10, respectively [14]; hence, the characterization of PGs is generally very complicated.

Hyperbranched PGs (HBPGs) can be sorted into two categories based on their molecular weight and, consequently, their applications: low molecular weight HBPGs (Mw < 2500 Da) and high Mw HBPGs with Mw of 24,000–700,000 Da [15,16].

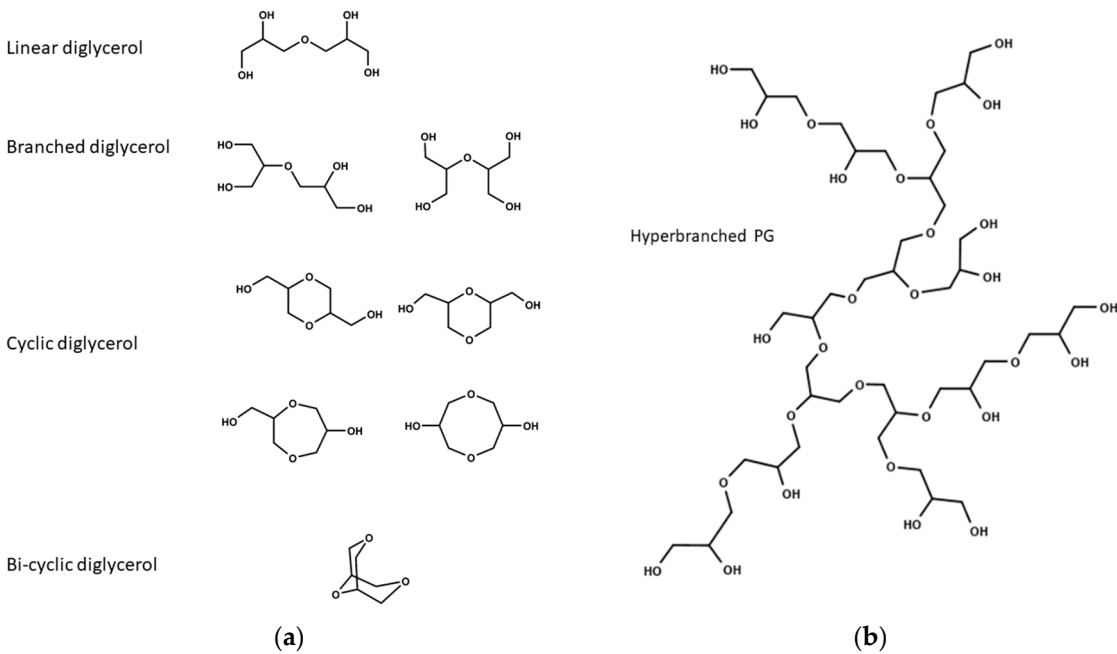

**Figure 1.** Different structures of: diglycerol including linear, branched and cyclic (**a**) (reproduced from [14]); and a hyperbranched PG (PG14) (**b**) (reproduced from [17]).

Biocompatibility and functionality of polyols are function of their molecular weight and of their number of hydroxyl end-groups. For example, hyperbranched PGs with a Mw of 5000 Da could possess up to 68 hydroxyl ends-groups, which makes it a highly functional material [18,19]. Therefore, HBPGs could have various applications, especially in nanobiotechnology and nanomedicine depending on their molecular weight [18].

Similarly, for PGs, as shown in Table 1, by increasing the PGs molecular weight, their properties such as viscosity and numbers of hydroxyl groups (i.e., their degree of functionalization) increase.

**Table 1.** Properties of linear polyglycerols commonly available on the market [11,12].

| Name (Abbreviation) | Molecular Weight (g mol$^{-1}$) | OH Groups | Viscosity (CTKS [1] @ 50 °C) | Density (g/cm$^3$) | Hydroxyl Value (mg KOH/g) [2] |
|---|---|---|---|---|---|
| Glycerol (G) | 92 | 3 | 45 | 1.256 | 1830 |
| Diglycerol (PG2) | 166 | 4 | 287 | 1.279 | 1352 |
| Triglycerol (PG3) | 240 | 5 | 647 | 1.2646 (40 °C) | 1169 |
| Tetraglycerol (PG4) | 314 | 6 | 1067 | 1.2687 (40 °C) | 1071 |
| Pentaglycerol (PG5) | 388 | 7 | 1408 | - | 1012 |
| Hexaglycerol (PG6) | 462 | 8 | 1671 | - | 970 |
| Heptaglycerol (PG7) | 536 | 9 | 2053 | - | 941 |
| Octaglycerol (PG8) | 610 | 10 | 2292 | - | 920 |
| Nonaglycerol (PG9) | 684 | 11 | 2817 | - | 903 |
| Decaglycerol (PG10) | 758 | 12 | 3199 | - | 880 |
| Pentadecaglycerol (PG15) | 1128 | 17 | 4893 | - | 846 |

[1] CTKS refers to centistokes. [2] The number of hydroxyl groups on a polyol is determined by reacting the polyol with acetic acid and then titrating with potassium hydroxide (KOH). The milligrams of potassium hydroxide required to neutralize one gram of the solution is called the hydroxyl value, as discussed in ASTM D4274-11.

As aforementioned, due to their properties, polyglycerols have found a wide range of applications in food, cosmetics, plastics and other industries. These applications obviously depend on their structure:

Linear polyglycerols are widely used in food industry as emulsifiers upon conversion to fatty acid esters, due to their specific physical characteristics, including a clear appearance at melting point, a desirable Gardner color, a mild odor and a bland taste [12,20,21]. Approximately 500,000 metric tons of polyglycerol fatty esters are produced worldwide. Sales in the European Union and the United States are estimated at EUR 200–300 million and USD 225–275 million, respectively [22]. Note that local regulations impose certain degrees of polymerization to the food industry; for instance, according to the EU law, PGs for food applications should not contain polymers containing more than seven condensed glycerol molecules (PG7) [2]. In contrast, PG2–PG10 have the U.S. Food and Drug Administration (FDA) approval [23]. Linear polyglycerols are known as being structurally similar to polyethylene glycol (PEG), which is produced from petrol-sourced ethylene oxide. Compared to PEG, polyglycerols have a higher water-solubility and even a slightly better biocompatibility, which make them a promising and renewable alternative to PEG [24,25].

Diglycerol (PG2) is widely used in cosmetic formulations as a solvent for fragrances. This short chain PG is non-toxic and has similar properties compared to that of glycerol but with a lower volatility due to its higher molecular weight. It also has a higher refractive index than glycerol, which is suitable for clear gels. A mixture of diglycerol with menthol also enhances the evaporation and impacts flavor and longevity in products such as toothpastes and mouthwashes compared to mixtures with glycerol [2,12]. Moreover, PG2 could have application in the cosmetics industry as emulsifier upon

conversion to polyglycerol esters to formulate oil/water emulsions for lotions, sunscreens creams and hair care products, as reported in [24] for Evonik Co. PGs with higher degree of polymerization (>PG3) are also attracting interest of cosmetics manufacturers as UV-absorbing polymers. For instance, mixture of polyglycerols containing PG3–PG12 (free of PG2 and its cyclic forms) with an average molecular weight higher than 500 Da has been utilized to formulate sunscreens and other related products [26].

High molecular weight polyglycerols, with high degree of branching and high functionality, are of interest in the plastic industry. For instance, hyperbranched polyglycerols have already found applications as surfactants in the plastic industry for treating lithographic printing plates and as organic solvents in aqueous inkjet-printing inks to prevent paper deformation. HBPGs can also be used as performance additives in water-based printing inks due to their higher solubility in water compared to polyesters [13,23].

Both linear and hyperbranched polyglycerols have the potential to be utilized in biomedical applications such as drug delivery, ophthalmic sealant for corneal wounds and anti-bacterial or anti-inflammatory agents [27]. However, polyglycerols must obviously meet drastic criteria to be used in biomedical applications such as high purity (they must be cyclic compounds-free and have a low coloration [2]).

As the physicochemical properties of polyglycerols can be varied by tuning their structure and degree of polymerization, PGs could have many other interesting applications. For instance, diglycerol can be also used as an oxygenated fuel additive [28]. PG3–PG6 have already been used as effective antifoam agents in many industries including food, pharmaceutical, cosmetic, paper, paints and coatings by partial conversion to polyglycerol esters [29,30]. PG10–PG20 can be used as foam stabilizers for oil fields and coal mines by mixing with non-ionic surfactants such as ammonia oxide [31].

## 3. Industrial Routes for Polyglycerols Production

Nowadays, polyglycerols manufacturers focus on two methods of production (Figure 2):

(a)    Basic hydrolysis of epichlorohydrin and glycerol
(b)    Direct polymerization of glycerol in the presence of a strong homogeneous base

**Figure 2.** Industrial routes for polyglycerol production: (**a**) Epicerol® process, with epichlorohydrin from glycerol prior to basic hydrolysis of epichlorohydrin (*n* and *m* = 1–4); and (**b**) direct polymerization of glycerol over an alkaline homogeneous catalyst (*n* and *m* = 2–10) (reproduced from [8,32–34]).

The first method starts from hydrolysis of epichlorohydrin to glycidol under basic conditions, before the as-formed glycidol is reacted with glycerol or unreacted epichlorohydrin to form diglycerol and higher oligomers, as shown in Figure 2a [12]. In 2011, Solvay introduced a process for the glycerol-based production of epichlorohydrin, the so-called Epicerol® process, which involves the reaction of glycerol with hydrochloric acid in the presence of Lewis acid catalysts followed by alkaline hydrolysis, as described in Figure 2a [32,33,35]. However, the overall relative low selectivity; the formation of high amounts of chloride salts (by-products); the need for many intermediate

steps, further separation and purification steps [33,35,36]; and most importantly the highly toxic and carcinogenic raw material (epichlorohydrin) are strong disadvantages of this method [9]. In addition, due to side reactions such as dehydration and oxidation reactions undesired colored products—leading to low product quality—is also an issue [12]. Besides, from an industrial point of view, such a process requires an explosion-proof facility, which represents a high capital investment [37].

In the second route (Figure 2b), the polymerization of glycerol is carried out using a strong homogeneous basic catalyst such as KOH or NaOH [20,21]. However, due to the use of the strong base, cyclic polyglycerols could form and often lead to degradation reactions forming byproducts with a dark color or a strong odor making them non-edible. Thus, bleaching agents such as hydrogen peroxide and sodium hypochlorite must be employed to improve the physical properties of the produced PGs [20], which is definitely not an environmentally-friendly solution. Using strong alkaline catalyst, such as NaOH and KOH as homogenous catalysts to directly convert glycerol to PGs, has other disadvantages: (i) the too fast reaction causes difficulty to control the degree of polymerization; (ii) the homogeneous catalyst cannot be recovered from the reaction medium and hence be reused; and (iii) strong bases induce corrosion issues of the industrial equipment.

Concerning the polymerization degree, PGs obtained by the "epichlorohydrin" route typically do not contain glycerol anymore. For instance, PG3 manufactured by Solvay contains a minimum of 80% di-, tri- and tetraglycerol, which is a mixture of linear, branched and cyclic forms, compared to a PGs manufactured by a "catalytic glycerol polymerization" route, which exhibit broader PGs' distributions [38] (Figure 3).

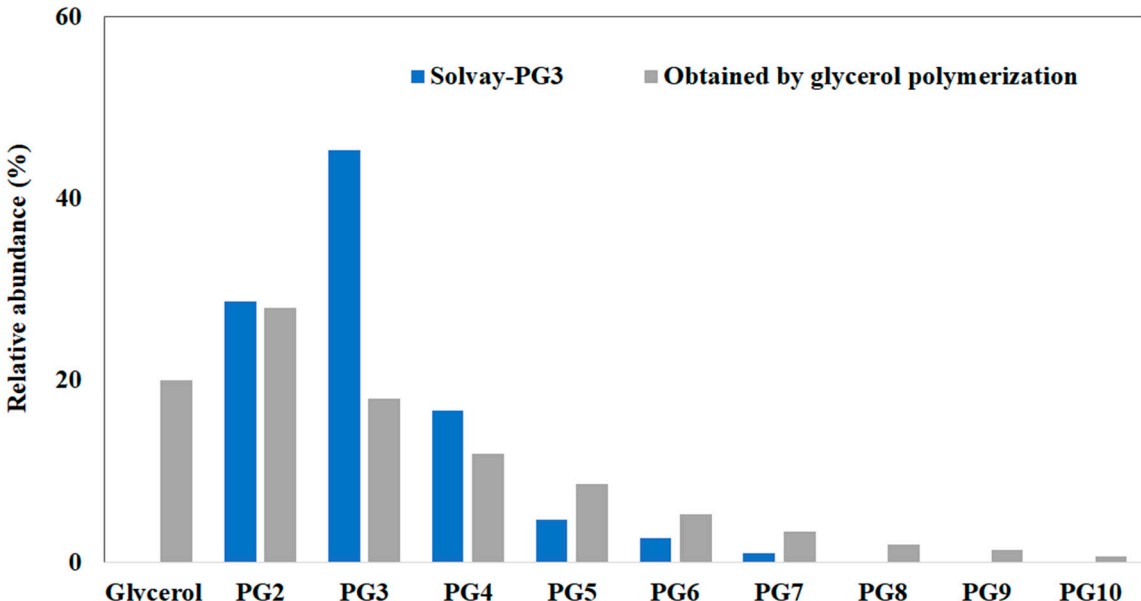

**Figure 3.** Comparison of polyglycerols distributions of Solvay-PG3 and polyglycerols obtained from catalytic polymerization of glycerol (average molecular weight = 250 Da) (reproduced from [38]).

The relative abundance in Figure 3 are described by Equation (1):

$$\text{Relative abundance (\%)} = 100 \times \frac{\text{amount of glycerol or desired PG}}{\text{total amount of glycerol and PGs}} \tag{1}$$

It should be noted that the quantification of PGs is obtained by analytical methods such as gas chromatography (GC) and high-performance liquid chromatography (HPLC), which do not need prior purification. However, if purified PGs are required, this can be done by distillation of the reaction mixture, as described by Babayan and Lehman [20].

## 4. Catalytic Systems

Several types of catalysts have been reported in the literature to polymerize glycerol to PGs, including strong acids [39–41], alkali-modified zeolites [42,43], bulk or supported alkaline oxides [3,5, 28,44], alkaline carbonate and hydroxide [21,28,45,46] and impregnated basic mesoporous solids [47]. Among all these catalysts, alkaline homogeneous and heterogeneous catalysts gave better performances than acid catalysts in terms of selectivity to PGs [9,41]. Thus, herein, this review is focused on homogeneous and heterogenous basic catalysts.

### 4.1. Homogenous Catalysis

Historically, the first route used to produce PGs from glycerol was homogenously-catalyzed by KOH or NaOH at 200–275 °C under $CO_2$ or $N_2$ at atmospheric or reduced pressure. Under such conditions, and in the presence of these alkaline catalysts, a wide range of polyglycerols (PG2–PG35) was produced [11,20,48].

Further, Garti et al. [46] screened the performance of several homogeneous catalysts at 260 °C under inert atmosphere. After 4 h of reaction in the presence of 2.5 mol% catalyst, their activity, based on mole percent of formed water per glycerol, followed the order: $K_2CO_3$ (94%) ~ $Li_2CO_3$ (94%) > $Na_2CO_3$ (92%) > KOH (79%) > NaOH (78%) > $CH_3ONa$ (71%)> LiOH (61%). Under these conditions, both the solubility of the catalysts in the reaction mixture and the catalyst basicity influence the reaction rate. While hydroxides are stronger bases than carbonates, $K_2CO_3$ was a better catalyst than KOH mainly due to a better solubility of the carbonate in glycerol. For the same reason, the oxides exhibited a lower activity. For instance, the rate of glycerol polymerization (based on mole percent of formed water per glycerol) was lower in the presence of CaO (7%) than $Ca(OH)_2$ (69%), as shown in Figure 4. It should be noted that in some works, oxides and hydroxides of alkali metals were considered as homogeneous catalysts because of their partial dissolution in glycerol, while they are not totally homogenous and could thus have dual catalytic roles by actually catalyzing the reaction both homogenously and heterogeneously. This point is discussed in more detail in Section 4.3.

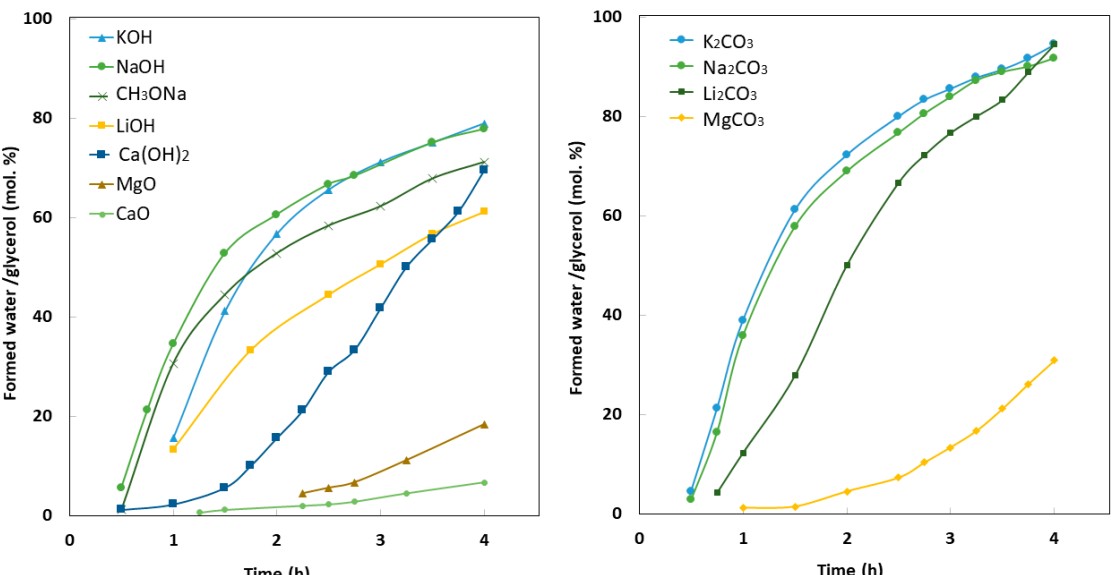

**Figure 4.** Rate of polymerization reactions based on mole of water formed during the reaction, using carbonates and hydroxides as catalysts (based on data presented in [46]).

However, it should be noted that the authors have not analyzed the obtained products; thus, the effect of the type of catalyst and of temperature on the polymerization degree is not clear. Similarly, Babayan and Lehman [20] used KOH and NaOH to produce mainly pentaglycerol. The reaction was carried out at 260 °C in the presence of 2 wt.% of catalysts under vacuum. Although the reaction was

stopped when the collected water reached the theoretical value of four moles, the obtained polymers were not identified and the selectivity to PGs was therefore not clearly determined. These catalysts were also applied for esterification of polyglycerols with oils to produce polyglycerols esters [20]. Charles et al. [49] also reported glycerol polymerization in the presence of 2 wt.% $Na_2CO_3$ and NaOH at 240 °C. After 9 h of reaction, glycerol conversions of 76% and 63% with selectivities of 93% and 99% to di- and triglycerol were, respectively, obtained. These results are in good agreement with those presented by Garti et al. [46], where carbonate was more active than hydroxide.

Calcium hydroxide was also used to produce linear polyglycerols by Lonza Co. [21]. The reaction was performed under vacuum at 230 °C and in the presence of only 0.1 wt.% of catalyst, which made it totally homogenous under such conditions. After 15 h of reaction, 57% of glycerol conversion was obtained with 87% of selectivity to di- and triglycerol. The maximum degree of polymerization observed was PG6. Cs salts including $CsHCO_3$, $Cs_2CO_3$ and CsOH were also used for glycerol conversion to diglycerol. The reactions were carried out at a temperature of 260 °C under atmospheric pressure with a Cs concentration of 1.85 mmol Cs/mol glycerol. These catalysts release different anions in the reaction media, however, resulting in the same selectivity to diglycerol (75%) when the glycerol conversion was 50% [45]. Nosal et al. [50] also reported glycerol polymerization in the presence of LiOH as a catalyst aiming at producing only di- and triglycerols by tuning the operational conditions. After 7 h of reaction, glycerol conversions of 24% and 80% with selectivities of 100% and 68% to di- and triglycerol were actually obtained in the presence of 0.1 wt.% LiOH at 230 °C and 260 °C, respectively.

Table 2 summarizes the different results discussed in this section.

**Table 2.** Summary of the conditions and performances obtained for glycerol polymerization over homogenous catalysts.

| Catalyst | Temp (°C) | Cat (wt.%) | Time (h) | Reaction Conditions | Glycerol Conversion (%) | Selectivity in PGx | Ref |
|---|---|---|---|---|---|---|---|
| LiOH | 240 | 2 | 6 | Continuous $N_2$ flow using Dean–Stark system to condensed water | 100 | $S_{PG2}$: 20% | [51] |
| LiOH | 230 260 | 0.1 | 7 | Low $N_2$ flow using Dean–Stark system to condensed water | 24 80 | $S_{PG2-3}$: 100% $S_{PG2-3}$: 68% | [50] |
| KOH NaOH | 260–280 | 2–4 | 1–4 | Continuous $N_2$ flow to remove formed water | 90–95 | PG10-25 | [52] |
| $CsHCO_3$ $Cs_2CO_3$ CsOH | 260 | 0.4 0.7 0.3 | 8 | Under atmospheric pressure | 64 71 75 | $S_{PG2}$: 23% $S_{PG2}$: 39% $S_{PG2}$: 32% | [45] |
| $Ca(OH)_2$ | 230 | 0.1 | - | Under vacuum (200 mmHg) | 57 | $S_{PG2-3}$: 87% | [21] |
| $Na_2CO_3$ NaOH | 240 | 2 | 9 | - | 76 63 | $S_{PG2-3}$: 93% $S_{PG2-3}$: 99% | [49] |
| CsOH | 260 | 2 | 4 | Under $N_2$ atmosphere using Dean–Stark system to condensed water | 90 | $S_{PG2-3}$: 63% | [47] |
| KOH NaOH | 260 | - | - | Continuous $N_2$ flow to remove formed water | 50–100 | - | [20] |

Besides the fact that, under basic homogenous catalytic reactions, the reaction rate is high, the lack of control of the glycerol polymerization degree leading to low selectivity and the impossibility to separate the catalyst from the reaction medium are still issues to be solved [12].

## 4.2. Heterogeneous Catalysts

In the last two decades, attention on glycerol polymerization over solid catalysts has been constantly growing, mainly because of the ease of separation by filtration or centrifugation after reaction, but also because of their higher selectivity [13].

Glycerol polymerization over basic solid catalysts has been investigated over many kinds of materials such as alkali-modified zeolites [42,43], impregnated basic mesoporous solids based on the MCM-41 family [47], bulk or supported alkaline earth metal oxides [5,6] or MgAl mixed oxides derived from layered double hydroxides [4,53]. Ruppert et al. [5] studied the catalytic effect of alkaline earth metal oxides including BaO, SrO, CaO and MgO with low specific surface areas (<5 m$^2 \cdot$g$^{-1}$) to produce mainly di- and triglycerol, while preventing the formation of higher oligomers by performing the reactions at relatively low temperatures (220 °C) for 20 h of reaction time. The order of glycerol conversion was reported as follows: BaO ≈ SrO (80%) > CaO (60%) >> MgO (<10%). This order is in accordance with the basicity of these earth metal oxide catalysts, namely: BaO > SrO > CaO > MgO. In the following, the same authors tuned the surface basicity and Lewis acidity in CaO catalysts by using CaO from different sources including commercial CaO, Ca(OH)$_2$ and calcium nitrate, of which the physicochemical properties differ. They reported that the CaO sample which possesses the stronger Lewis acidity exhibited a higher glycerol conversion. However, Lewis acidity does not seem to be the sole parameter that could influence polymerization, as MgO which exhibits the strongest Lewis acidity among this series of catalysts has the lowest catalytic activity, whereas BaO and SrO with no Lewis acidity the highest catalytic activities.

As high selectivity towards short chains PGs, including di- and triglycerols, was of interest, research groups started to tune and design porous catalysts. In fact, the pore diameter can orientate the selectivity to particular PGs isomers through the so-called "shape selectivity" concept. For instance, to favor diglycerol selectivity, the catalyst pores should have a size larger than 0.712 and 0.753 nm to induce formation of αα′ and ββ′ diglycerol isomers, respectively, and to enable the access of glycerol with a kinetic diameter of 0.515 nm (calculated by DFT) [12].

Krisnandi et al. [42] designed Na-modified microporous zeolites to enhance the shape selectivity toward diglycerol formation. The zeolite structure could induce suppression of the formation of bulky products such as trimer and higher oligomers, while diglycerol could form inside the microporous structure. Thus, NaX, NaY and NaBeta were used as catalysts in the polymerization reaction at 260 °C. For 8 h reaction time, NaX was the most active catalyst, achieving 50% glycerol conversion with 80% selectivity to diglycerol, followed by NaY and NaBeta, which led to 20% and 10% glycerol conversion, respectively, with 100% selectivity to diglycerol. However, between 4 and 6 h reaction time, Na started to leach out of the zeolite, and, after 24 h, the zeolite structure was destroyed. Thus, the high conversion observed in the presence of NaX was explained by its low stability in the presence of water which resulted finally in a homogenous catalytic reaction. Furthermore, there are other limitations of using microporous zeolite catalysts for glycerol polymerization reactions including mass transfer limitations in internal pores which could limit the shape selectivity and also the low selectivity to PGs [10]. For instance, Ayoub et al. [54] used a lithium-exchanged zeolite Y with average pore size of 3.1 nm at 240 °C to polymerize glycerol and obtained 99% glycerol conversion with only 72% selectivity to polyglycerols after 8 h of reaction. They explained an unexpected low selectivity to diglycerol (21%) by mass transfer limitation within internal pores while higher PGs formed at the external surface of the catalyst. However, the high glycerol conversion was probably caused by the leaching of Li into the reaction media, but this was not considered in this study.

On the other hand, Gholami et al. [55,56] also investigated glycerol polymerization over a Ca$_{1.6}$La$_{0.4}$Al$_{0.6}$O$_3$ mixed oxide catalyst. They obtained a glycerol conversion of 98% with 53% diglycerol selectivity after 8 h of reaction at 250 °C. They concluded that larger pores (average pore size of 18 nm) lead to higher diglycerol selectivity by facilitating the diffusion of glycerol molecules into the mesopores of the catalyst. However, in the same group, a glycerol conversion of 91% with 47% diglycerol selectivity was obtained in the presence of 20% Ca$_{1.6}$La$_{0.6}$/MCM-41 with a pore diameter of

1.4 nm and in the same reaction conditions [57]. Thus, the pore size of the catalysts had not such a significant effect on diglycerol selectivity. This may be due to the fact that Ca and Li could leached into the reaction media and that then the reaction is homogeneously catalyzed [57].

García-Sancho et al. [3] designed a porous solid catalyst based on Mg/Al mixed oxides derived from hydrotalcites and tuned the shape-selectivity of glycerol polymerization. The porous structure of these compounds was expected to promote the formation of desired short chain PGs. The reactions were carried out under batch conditions at 220 °C for 24 h. The authors reported only the formation of di- and triglycerol, but the conversion was relatively modest: the highest conversion was 50% over MgAl-Na at near 80% selectivity toward diglycerol, while 100% selectivity was obtained over MgAl-Urea, but with only 18% glycerol conversion. They concluded that the lower is the pore diameter of the catalyst, the higher is the diglycerol selectivity. In the same group, MgFe mixed oxides were developed in order to introduce heterogeneous acid sites ($Fe^{3+}$) to MgO. Polymerization of glycerol was performed over magnesium iron hydrotalcites at 220 °C for 24 h in batch reaction conditions. In these conditions, high diglycerol selectivities of 90% and 100% were obtained over $MgFeO_4$ and $MgFeO$, with a glycerol conversion of 41% and 21%, respectively. The authors concluded that the acidity, basicity and porosity of the catalysts contributed to their performances. Note that acrolein formation due to the presence of acid sites in these catalysts was not reported [58].

Besides, Pérez-Barrado et al. [59] also investigated the influence of acidity and basicity of different MgAl and CaAl layered double hydroxides (LDHs) on short chain PGs' selectivity. They observed that catalysts with higher acidity (higher number of acidic sites) showed higher conversion (75–96%) and low selectivity to PGs (88% selectivity to acrolein). In contrast, catalysts with less acid sites and higher amounts of medium strength basic sites resulted in lower conversion (24%) and higher selectivity towards di- and triglycerol (100%).

Similar results were reported by Sangkhum et al. [60], who studied glycerol polymerization in the presence of a Ca-Mg-Al mixed catalysts derived from layered double hydroxide. They reported the highest diglycerol selectivity of 78.3% at a relatively low glycerol conversion of 40.4% when the reaction was performed at 220 °C and in presence of 3 wt.% catalyst for 24 h.

More recently, Barros et al. developed new catalysts to obtain short chain PGs. Calcined dolomite, mixed carbonate of calcium and magnesium ($CaCO_3 \cdot MgCO_3$) and calcined eggshell were utilized to catalyze the polymerization reaction. In the presence of dolomite, the glycerol conversion was 77% with selectivities of 51% and 3% to di- and triglycerol, respectively, at 220 °C for 24 h of reaction [7]. Under the same reaction conditions, and in the presence of calcined eggshell, the glycerol conversion of 85% with 40% selectivity of diglycerol were obtained [3]. Over both catalysts, increasing the reaction temperature led to an increase in conversion but at the expense of diglycerol selectivity.

In contrast to these studies where the product of interest was diglycerol, Bruijnincx's team synthesized a new catalyst based on CaO in order to obtained higher PGs (>PG4). They used carbon nanofibers (CNF) as a support for CaO to yield larger specific surface areas while the active species (CaO) was mostly unaffected by the presence of the CNF as the interactions between CNF as a support and active phase were low. The highest conversion was obtained in the presence of 14% CaO/CNF with a glycerol conversion was 76% with selectivities to di-, tri-, tetra- and higher oligomers of 38%, 20%, 10% and 5%, respectively, at 220 °C after 24 h of reaction [6,44,61].

Table 3 summarizes the results of the studies on glycerol polymerization over heterogeneous catalysts. However, with respect to the problem of leaching and hydrothermal stability of the catalysts, one cannot clearly conclude if all the reported results on heterogeneous catalysts do not finally exhibit a contribution of homogeneous catalysis from leached or dissolved species. Hence, we discuss this issue in the next section.

**Table 3.** Summary of the conditions and performances obtained for glycerol polymerization over heterogeneous catalysts.

| Catalyst | Temp (°C) | Cat (wt.%) | Time (h) | Reaction Conditions | Glycerol Conversion (%) | Selectivity to PGx | Ref |
|---|---|---|---|---|---|---|---|
| Ca-MgAl LDH | 220 | 3 | 24 | Under $N_2$ flow using Dean–Stark system to collect water | 40.4 | $S_{PG2}$: 78.3% | [60] |
| Dolomite (mixed oxide CaO-MgO) | 245 | 2 | 24 | Under $N_2$ atmosphere using Dean–Stark system to condense water | 90 | $S_{PG2}$: 23% $S_{PG3}$: 22% | [7] |
| Calcined eggshell | 220 245 | 2 | 24 | Under $N_2$ atmosphere | 85 100 | $S_{PG2}$: 35% $S_{PG3}$: 15% | [3] |
| Duck bones | 240 | 2 | 12 | Under $N_2$ atmosphere using Dean–Stark system to condense water | 99 | - | [51] |
| 14wt.% CaO/CNF | 220 | 0.46 * | 24 | Under Ar gas flow using Dean–Stark system to condense water | 76 | $S_{PG2}$: 40% $S_{PG3}$: 15% | [44] |
| MgAl-LDHs–CaAl-LDHs: cHT1 | 235 | 2 | 24 | Under $N_2$ atmosphere using Dean–Stark system to condense water | 24 | $S_{PG2-3}$: 100% | [59] |
| MgAl-LDHs–CaAl-LDHs: cHT2 | 235 | 2 | 24 | Under $N_2$ atmosphere using Dean–Stark system to condensed water | 96 | $S_{PG2-3}$: 12% $S_{acrolein}$: 88% | [59] |
| $MgFeO_4$ | 220 | - | 24 | Under $N_2$ atmosphere using Dean–Stark system to condense water | 41 | $S_{PG2}$: 90% $S_{PG3}$: 10% | [58] |
| $Ca_{1.6}La_{0.4}Al_{0.6}O_3$ | 250 | 2 | 8 | Under $N_2$ atmosphere using Dean–Stark system to condense water | 96.3 | $S_{PG2-3}$: 86% | [55] |
| MgAl-Na | 220 | 2 | 24 | Under $N_2$ atmosphere using Dean–Stark system to condensed water | 50 | $S_{PG2}$: 85% $S_{PG3}$: 15% | [4] |
| $Ca(OH)_2$ $CaCO_3$ | 140 | 2.4 | ~6 | Under reduced pressure using Dean–Stark system to condensed water | 12 11 | - | [40] |
| $CaO/Ca_{12}Al_{14}O_{33}$ | 200–250 | - | 24 | Under $N_2$ atmosphere using Dean–Stark system to condensed water | - | Linear PGs | [62] |
| Zeolite NaX Zeolite NaY Zeolite Na Beta | 260 | 2 | 24 | Under Ar atmosphere | 100 80 50 | $S_{PG2}$: 15% $S_{PG2}$: 58% $S_{PG2}$: 90% | [42] |
| MgO BaO-SrO CaO | 220 | 2 | 20 | Under Ar atmosphere using Dean–Stark system to condensed water and Ice trap for acrolein | 10 80 > 80 | $S_{PG2-3}$: 90% | [5] |
| $Cs_{25}Al(Si/Al:20)$ | 260 | 2 | 20 | Under $N_2$ atmosphere using Dean–Stark system to condensed water | 80 | $S_{PG2}$: 62% $S_{PG3}$: 33% | [47,63] |
| Zeolite NaA Zeolite NaZ | 200–260 | | 22 | Under $N_2$ with a reflux condenser and water separator | 84.6 90.5 | $S_{PG2-3}$: 62% $S_{PG2-3}$: 52% | [43] |

* mol.% of Ca.

However, the possibility of acrolein production as a by-product (mainly in the presence of acid catalysts through glycerol double dehydration) and the instability of the catalysts due to the formation of water in which they are generally partially soluble are still issues to be tackled [40].

### 4.3. Partial Dissolution of Heterogeneous Catalysts

Partial dissolution of CaO-based catalysts generally occurs in the reaction medium and hence a subsequent homogeneous catalysis contribution makes the catalytic system more complex to understand. As a matter of fact, the formation of colloidal $Ca(OH)_2$ and the leaching of $Ca^{2+}$ in the reaction medium following Equation (2) has been often reported in glycerol polymerization [5,6,42,47].

$$2CaO + 2H_2O \leftrightarrow Ca(OH)_{2(l)} + Ca^{2+} + 2OH^- \qquad (2)$$

Ruppert et al. [5] measured total $Ca(OH)_2$ and $Ca^{2+}$ in the reaction medium by inductively coupled plasma (ICP) analysis. They further observed colloidal $Ca(OH)_2$ particles with a size of 50–100 nm by cryo-transmission electron microscopy (TEM). Their results show that the Ca concentration in the reaction medium reached 3–6 wt.% after only 2 h of reaction when using calcined CaO as catalyst. The authors concluded that the formation of colloidal $Ca(OH)_2$ and leached $Ca^{2+}$ could contribute to the

polymerization reaction beside solid phase catalyst in the reaction medium. In addition, the colloidal $Ca(OH)_2$ particles could not be separated from the reaction mixture and caused a non-transparent mixture (low quality product) and progressive deactivation upon recycling.

Nieuwelink [6] studied the Ca leaching as function of the temperature for 14 wt.% CaO/CNF with dynamic light scattering (DLS) and conductivity measurements at 100, 150 and 220 °C. On one hand, their results indicate that at 100 °C no colloid formation was observed, but a large amount of dissolved $Ca^{2+}$ ions, with a conductivity of 1118.7 µS/cm, into the glycerol mixture was observed which caused the saturation of glycerol. On the other hand, at higher temperatures of 150 and 220 °C, more colloid formation and less $Ca^{2+}$ free species were observed. This could be linked to the fact that the polymerization reaction at higher temperatures released more water, hence promoting the formation of colloidal $Ca(OH)_2$ according to Equation (1). Further, in the same group, Kirby et al. [44] reported that the active phase derived from CaO/CNF catalyst might be the colloidal $Ca(OH)_2$ particles and $Ca^{2+}$. The amounts of Ca and Mg leaching from dolomite were also determined by ICP-OES during the glycerol polymerization reaction by Barros et al. [7]. According to these authors, 49% Ca and 4% Mg leached into the medium at 245 °C after 24 h of reaction. They noted that a homogenous contribution of CaO could play a role on glycerol conversion, as a significant drop in conversion was observed when dolomite was isolated and reused for the next test.

Cs leaching has also been reported during glycerol polymerization reactions in the presence of Cs-modified zeolites [47]. A lower conversion with reused-$Cs_{25}Al(20)$ was observed in comparison with the initial catalyst (60% vs. 80%), while both catalysts had similar selectivities to di- and triglycerol. Similarly, for Na-exchanged zeolite catalysts, Na leaching out of the zeolite framework during the first 4–8 h of reaction followed by total dissolution of the solid catalysts at 260 °C and after 24 h reaction time was also reported by [42].

Ca and Li leaching also caused a very high glycerol conversion (91%) after 8 h of reaction when 20% $Ca_{1.6}La_{0.6}$/MCM-41 was used as catalyst at 250 °C, suggesting a homogenously-catalyzed reactions [57].

To conclude, several so-called "heterogeneously-catalyzed" reactions were in fact carried out in totally or partially homogeneously-catalyzed condition, either by dissolution of the catalysts itself or by leaching of active species into the reaction medium. However, the homogeneous contributions are not very clear, as most often they were not closely studied or even considered. Most of the time the materials used are still classified as solid heterogeneous catalysts.

## 5. Mechanism of Glycerol Polymerization over Alkaline Catalysts

The mechanism of polymerization of glycerol over alkaline catalysts comprises two steps: (i) deprotonation of a hydroxyl group as a nucleophile; and (ii) attack of the as-formed alkoxy anion (glyceroxide ion) on a carbon of another glycerol molecule. However, due to the fact that hydroxyl groups are poor leaving groups, different mechanisms have been proposed to explain the process. For instance, for solid alkaline catalyst such as CaO, MgO, BaO and SrO [5], it is presumed that a high reaction temperature is required for the formation of PGs by facilitating the leaving of hydroxy group; another assumption is that Lewis acid sites take part in the mechanism through activation of a hydroxyl group as a leaving group (Figure 5). However, as aforementioned, the contribution of Lewis acidity in PGs reaction mechanism cannot explain the lowest activity of MgO and the highest activity of BaO and SrO. Moreover, for other catalysts that possess Lewis acid sites such as CaAl layered double hydroxides, the reaction favored acrolein formation (Table 3).

**Figure 5.** Reaction mechanism for glycerol polymerization over earth metal oxides catalysts as proposed by [5].

Salehpour and Dubé [40] proposed a mechanism for catalysis over $Ca(OH)_2$ suggesting that a protonated glycerol is a better nucleophile that could then attack a second molecule of glycerol, as a weak electrophile, on one of the primary or secondary alcohol groups (Figure 6). This mechanism could explain the slower rate of alkaline-catalyzed reaction compared to the acid-catalyzed one, where glycerol becomes a better electrophile due to $^+H_2O$-C bond. Thus, to break the C-O bond and achieve a higher reaction rate, a higher reaction temperature should be applied for alkaline-catalyzed reaction. In this mechanism, calcium ions are also involved by coordinating to the oxygen atoms of glycerol. This coordination, as the so-called *"pseudo-bond"*, could cause the carbon–oxygen bond of a glycerol molecule to lengthen and thus induce lowering of the energy of the transition state complex. In this mechanism, the $Ca(OH)_2$ should be in homogeneous state to access Ca ions to be involved in the mechanism. Besides, this mechanism could also be assumed for CaO, by Ca leaching into the reaction medium, as aforementioned. Thus, this suggests that the actual mechanism with calcium oxide and hydroxide is actually homogeneous.

**Figure 6.** Reaction mechanism for glycerol polymerization with $Ca(OH)_2$ as a catalyst, with contribution of Ca in the mechanism, as proposed in [40].

Martin and Richter [12] suggested that a homogenously alkaline-catalyzed reaction follow a SN2 mechanism, as shown in Figure 7. Similarly, the protonated glycerol molecule (glyceroxide ion) is a nucleophilic species that attacks the hydroxyl group of glycerol, and then forms a water molecule. However, the corresponding cations does not contribute to this mechanism.

**Figure 7.** Reaction mechanism for homogenously catalyzed glycerol polymerization reaction, as proposed in [12].

On the other hand, Ionescu and Petrović [8] presumed that glyceroxide ion could transform to glycidol by intramolecular nucleophilic substitution. Then, glycidol, as an intermediate, reacts with the hydroxyl groups of glycerol to form a dimer (Figure 8). This is similar to the classical mechanism for formation of polyglycerols by ring opening of glycidol with a fast reaction rate at room temperature (Figure 1a) [12,64]. However, while it proposes the well-known glycidol ring opening mechanism to produce PGs, it cannot explain the roles of oxides as heterogeneous alkaline catalysts and their slower reaction rate.

**Figure 8.** Reaction mechanism for alkaline-catalyzed reactions by glycidol formation, as proposed in [8].

According to all these mechanisms, glycerol polymerization reaction is subjected to a high activation energy, translated with the need of high temperatures to make the reaction possible. The operating conditions actually used are discussed in the next section.

## 6. Reaction Conditions

### 6.1. Temperature

According to the literature, temperature plays an important role in glycerol polymerization reactions catalyzed by both homogenous and heterogeneous catalysts. The studied temperature range for alkaline heterogeneously catalyzed reactions starts from 200 °C to a maximum of 270 °C, as presented in Table 3, keeping in mind that the boiling point of pure glycerol is 290 °C under atmospheric pressure [65].

Sangkhum et al. [60] studied the effect of the temperature in the polymerization of glycerol over Ca-MgAl mixed metal oxide as a catalyst. They performed the reactions at the temperatures of 200, 210, 220 and 230 °C and observed, as expected, a significant increase in glycerol conversion from 40% at 220 °C to 86% at 230 °C, while glycerol conversion was only 5% and 11% at the lower temperatures of 200 and 210 °C, respectively. They also reported a decrease in diglycerol selectivity from nearly 78% to 33% at 220 and 230 °C, respectively.

Gholami et al. [55] investigated the effect of temperature in the presence of $Ca_{1.6}La_{0.4}Al_{0.6}O_3$ as a catalyst. A significant increase in glycerol conversion from 28.1% at 220 °C to nearly 96% at 260 °C after 8 h of reaction was reported, where the yield in di- and triglycerol was 11% and 88%, respectively. However, a drop in diglycerol yield (77%) was observed at higher temperature (260 °C), which was explained by the formation of higher polyglycerols (>PG3) or cyclics PGs.

Kirby et al. [44] also studied the effect of temperature on the polymerization of glycerol and the coloration of products in the presence of CaO-CNF. They performed tests at the temperatures of 180, 200, 220, 240 and 260 °C and observed glycerol conversions of ~10, ~35, ~55, 80 and 100%, respectively. Moreover, they concluded that the highest studied temperature (260 °C) gave rise to the formation of undesired products (dark color products) like cyclic dimers, acrolein and other dehydration products with a decrease in selectivity to shorter PGs (PG2 and PG3).

Similar results were reported by Barros' team, that observed an increase in glycerol conversion when increasing the temperature from 200 to 245 °C in the presence of an eggshell commercial CaO [3] and calcined dolomite [7]. For instance, the conversion increased from 10% at 200 °C to 80% and 100% at 220 °C and 245 °C, respectively. When diglycerol selectivity decreased, the formation of acrolein increased, and a darker and more viscous product was observed, as shown in Figure 9.

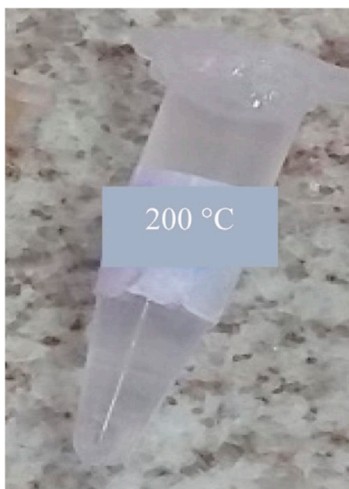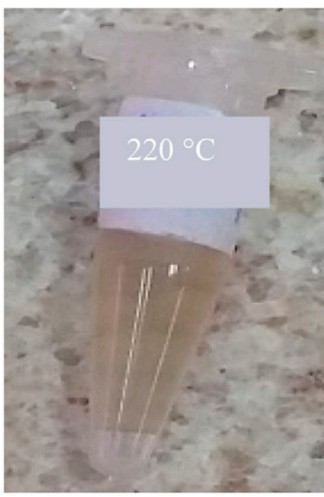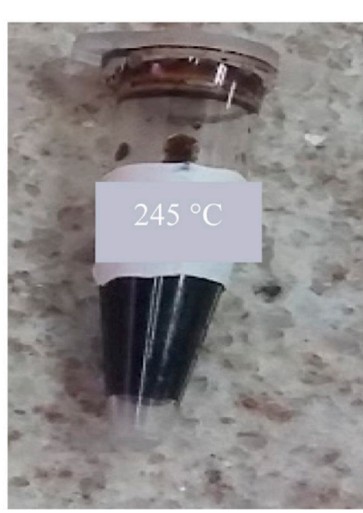

**Figure 9.** Effect of temperature on the transparency of the product of glycerol polymerization in the presence of calcined eggshell (CaO) (reprinted from [3], copyright 2020, Elsevier).

## 6.2. Effect of Catalyst Loading

Garti et al. [46] mentioned that increasing the amount of catalyst can accelerate the reaction rate, further affecting the overall degree of polymerization. Based on the amount of water formed during the reaction, 0.5–1 mol.% of NaOH as a catalyst were proposed for obtaining di- and triglycerols and 4–10 mol.% for higher degree of glycerol polymerization.

Sangkhum et al. [60] investigated the effect of Ca-MgAl mixed oxide amount on glycerol polymerization at 220 °C. According to their results, glycerol conversion increased to 20% and 40% in the presence of 2 and 3 wt.% of catalyst, respectively, whereas the glycerol conversion was 6% without catalyst. They reported that increasing the catalyst loading also promoted diglycerol selectivity from 40.4% to 78.3%, respectively (Figure 10). However, when the catalyst loading was further increased at 5 wt.%, the glycerol conversion and diglycerol selectivity declined, reaching 27.9% and 42.6%, respectively. They explained this phenomenon by the increase in glycerol conversion and in the quantity of formed water in the presence of higher amount of catalyst, which might cause back-scission of polyglycerol to glycerol via hydrolysis. However, this explanation is not very convincing since the role of water in an equilibrium reaction would be the same whatever the catalytic system and catalyst amount.

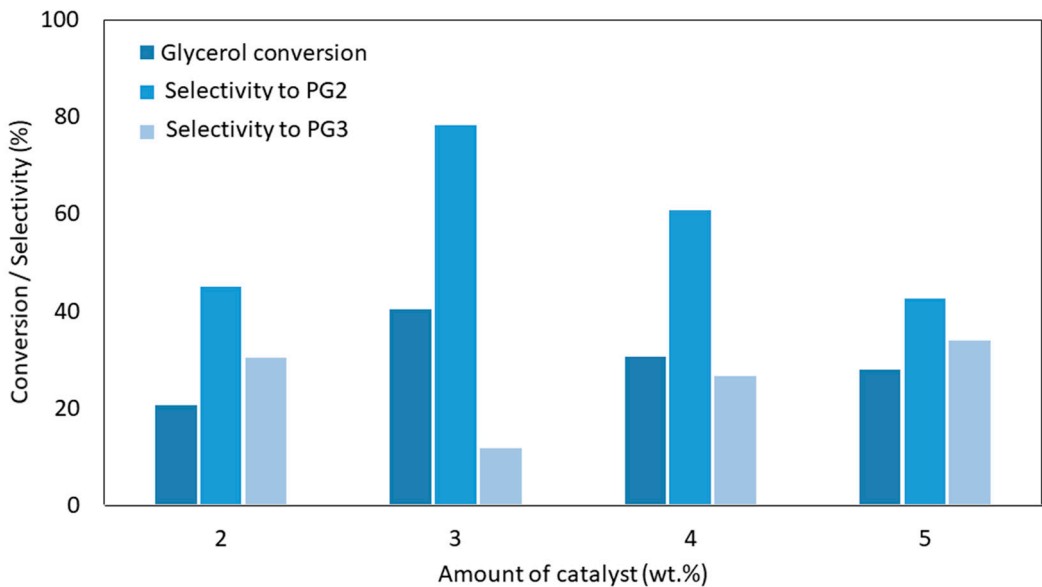

**Figure 10.** Catalytic conversion of glycerol over 7.5% Ca-MgAl mixed metal oxide at 220 °C for 24 h (reproduced from [60]).

Kirby et al. [44] studied the effect of CaO, Ca(OH)$_2$ and CaO/CNF loading on glycerol conversion to PGs. According to their results, increasing the amount of CaO from 0.33 to 3.28 mol.% (equivalent to 2 wt.% CaO) had no significant effect on glycerol conversion, while for CaO/CNF, an increase in the amount of catalyst from 0.16 to 0.46 mol.% induced an increase in conversion from 48% to 76%. They explained this result by homogenous contribution of CaO and Ca(OH)$_2$ in the reaction with complete dissolution of CaO in the reaction mixture at loadings up to 0.46 mol.% to form Ca(OH)$_2$ (from CaO) and Ca$^{2+}$ species, as previously discussed.

On the other hand, Barros et al. [3] reported an increase in the glycerol conversion from 34% to 85% by increasing the amount of calcined eggshell CaO as a catalyst from 0.5 to 2 wt.% while selectivity to di- and triglycerol decrease from nearly 55% to 43%.

## 6.3. Pressure

With respect to glycerol polymerization, Jungermann et al. [11] noted that using vacuum at high reaction temperature increased the glycerol conversion by eliminating the formed water from the reaction medium and lead to a narrowing of the range of PGs molecular weight. Although they did not mention any values for the applied vacuum and temperature in polymerization reaction, Lemke [16] reported the use of 230 °C and under vacuum of 200 mm Hg. However, the obtained results were not compared with results obtained at atmospheric conditions. In practice, in the case of solid catalysts, the polymerization reaction is mainly carried out under atmospheric pressure using a Dean–Stark apparatus with a reflux condenser to continuously remove water formed during the reaction (as mentioned in Table 3).

## 6.4. Atmosphere

Traces of oxygen lead to the formation of oxidation products. Therefore, to prevent parasite oxidation reactions, glycerol polymerization is preferentially carried out under an inert gas such as N$_2$ or Ar [46]. However, while recent studies have mostly been carried out under N$_2$ atmosphere (see Tables 2 and 3), a few studies were also carried out under CO$_2$ atmosphere. For instance, Garti et al. [46] used both N$_2$ and CO$_2$ atmospheres for glycerol polymerization and reported similar production of tetra- and penta-polyglycerol under both conditions. Wilson et al. [66] employed CO$_2$ as an inert gas to prevent oxidation for glycerol conversion to polyglycerol at 200 °C in the presence of potassium hydroxide. However, the reason behind applying CO$_2$ is not clear. We can suppose that

it might be linked to the $CO_2$ Lewis acidic nature, which could contribute to the reaction. However, the impact of $CO_2$ highly depends on temperature, solubility of $CO_2$ in glycerol, concentration of $CO_2$, pressure [67] and further formation of glycerol carbonate [68]. Moreover, many studies kept CaO under nitrogen or used freshly calcined CaO in order to prevent "deactivation" of catalyst by contact with ambient air (reactions with $CO_2$ and $H_2O$) and formation of $CaCO_3$ and $Ca(OH)_2$, respectively [5,6].

*6.5. Reaction Time*

Polymerization of glycerol in the absence of a solvent and in the presence of a heterogeneous catalyst is a slow reaction because the reactants must diffuse into the catalyst in a viscous reaction mixture [55]. Consequently, longer reaction times are needed to overcome diffusion resistance and reach higher degrees of polymerization. For instance, Kirby et al. [44] studied the composition of reaction mixture during 24 h at 220 °C in the presence of CaO/CNF. They observed an increase in formation of di- and triglycerol during the first 8 h of reaction, and then a progressive decrease of these compounds to the benefit of higher oligomers. Glycerol polymerization reactions catalyzed by solids generally take place within 24 h, as shown in Table 3.

Overall, glycerol self-condensation reactions are influenced by many factors and various possible mechanisms depending on the catalyst types, etc. As a consequence, the average degree of polymerization could be affected directly by several factors including conversion. Hence, in this case, using an equation such as Carothers equation would unfortunately not describe the real average polymerization degree, and assessment of the performances by finely analyzing and comparing each output parameter is needed.

**7. Conclusions**

In this review paper, polyglycerols are shown to be very interesting polyols with a wide range of structures and, accordingly, of applications, particularly in their ester forms for the cosmetics, biomedical and food sectors. Actually, PGs are water soluble, biocompatible and highly functional materials. While short chain PGs are of interest for food and cosmetic industries, higher degrees of polymerization are of high interest in plastic industry because they induce an increase in the possibility of PGs further functionalization by increasing the numbers of available hydroxyl groups. The current industrial methods to produce PGs, such as basic hydrolysis of epichlorohydrin, have many disadvantages, such as toxic and carcinogenic starting materials, low selectivity, formation of high amounts of salts, etc. Using strong alkaline catalyst, such as NaOH and KOH as homogenous catalysts to directly convert glycerol to PGs, has also several disadvantages such as the difficulty to control the degree of polymerization; the impossibility to recover the homogeneous catalyst from the reaction medium after reaction and hence to reuse it; and the corrosion issues of the industrial equipment in presence of strong bases.

Hence, in the last two decades, attention has been paid to develop a heterogeneously-catalyzed process for glycerol direct polymerization over alkaline solid catalysts due to their advantages over alkaline homogeneous and acidic catalysts, such as a better control of the degree of polymerization, the ease of catalyst separation and recycling, as well as being non-corrosive for the industrial equipment. Based on the literature review, alkaline earth oxides such as CaO, BaO and SrO are active catalysts for polyglycerols production and present a lower tendency to form acrolein (main by-products issued from glycerol double dehydration). For instance, CaO, SrO and BaO have shown glycerol conversion close to 80% together with 90% selectivity to di- and triglycerol [5]. Ninety-six percent of glycerol conversion with 86% selectivity to di- and triglycerol were also reported in the presence of $Ca_{1.6}La_{0.4}Al_{0.6}O_3$ [55]. Moreover, among alkaline earth oxides catalysts, CaO-based catalysts such as CaO, supported CaO on CNF, dolomite (CaO-MgO mixed oxides), calcined eggshell and duck bones have shown high glycerol conversion with relatively high selectivity to PG2 and PG3 and nearly zero acrolein formation. However, these catalysts generally had a high selectivity to PG2 and PG3 and low selectivity to higher PGs, probably because of the low reaction temperature (Table 3). More importantly, they were

unstable under polymerization reaction conditions mainly due to water formation during the reaction, which caused a partial dissolution of the catalyst and hence a homogenous contribution during glycerol polymerization reaction. This homogenous contribution has been reported by many groups in the forms of colloidal $Ca(OH)_2$ and $Ca^{2+}$ [3,5,6]. For instance, Barros et al. [3] reported that 49% of solid CaO-eggshell became homogeneous (i.e., dissolved in the reaction medium) after 24 h of reaction.

The proposed mechanism over $Ca(OH)_2$ [40] also suggested an homogeneously catalyzed reaction by Ca ion species, which could explain the roles of leached species in the PGs reaction; the only mechanism proposed for heterogenous alkaline catalysts [5] involved Lewis acid sites, and cannot explain the highest activities of BaO and SrO with no acidic sites.

Besides the catalyst, the operational conditions such as temperature, pressure, atmosphere and reaction time can play a very important role on the glycerol conversion and the degree of polymerization of glycerol. For instance, Sangkhum et al. [60] reported that the glycerol conversion is doubled when the temperature is increased from 220 to 230 °C. Further, Kirby et al. [44] and Gholami et al. [55] also reported a full conversion of glycerol at 260 °C and an increase the formation of higher polyglycerols (>PG3) and cyclic PGs. However, the effect of pressure and atmosphere has not been well studied in glycerol polymerization.

Thus, considering the high catalytic activity of CaO based catalysts and their wide availability, low cost and absence of toxicity in the case of catalyst leaching into the media, they could be viewed as promising for glycerol polymerization reactions. These alkaline catalysts could promote the polymerization of glycerol to shorter PGs, such as PG2 and PG3, at low temperature (200–230 °C) and induced the formation of higher degree of polymerization at higher temperatures (>230 °C). However, a key issue that has yet to be fully addressed is the effect of formed water in the system causing dissolution of the catalysts and homogeneous contribution in the glycerol polymerization reaction. Although the heterogeneously catalyzed reactions were performed in a Dean–Stark system to condense and collect water from the reaction mixture (Table 3), the formed water can still cause a partial dissolution of the catalysts. Researchers should therefore strive to design stable solid catalysts to polymerize glycerol and also investigate the homogenous contribution in the formation of PGs. Taking into account that the proposed mechanisms for PGs reaction are mostly for homogeneously catalyzed reactions [8,12,40], distinguishing the influence of homogenous and heterogenous catalysts on polymerization of glycerol would also help understanding the "true" reaction mechanism, which is still matter of debates.

**Author Contributions:** N.E. wrote the paper; S.P., F.D. and B.K. led the study, revised the paper and gave advice on interpretation of references to complete the review and its conclusions. All authors have read and agreed to the published version of the manuscript.

**Funding:** Negissa Ebadipour's PhD grant was co-funded by Centrale Lille and Région Hauts-de-France.

**Acknowledgments:** Chevreul Institute (FR 2638), Ministère de l'Enseignement Supérieur, de la Recherche et de l'Innovation, Région Hauts-de-France and European Regional Development Fund (ERDF) are acknowledged for supporting this work. The authors would like to thank Joël Barrault for providing us some PGs samples.

**Conflicts of Interest:** The authors declare no conflict of interest.

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
