# Peer review of "Alkaline-Based Catalysts for Glycerol Polymerization Reaction: A Review"

_catalysts, doi:10.3390/catal10091021_

Round 1

Reviewer 1 Report

This manuscript reviews the uses of homogeneous and heterogeneous alkaline catalysts for glycerol oligomerization/polymerization.  I do not recommend the publication of this manuscript because it has the following problems:

  • The subject has been reviewed this year by Chong et al. (please see reference 10: A review over the role of catalysts for selective short-chain polyglycerols and other value-added products).
  • The heterogeneous alkaline reviewed in this manuscript have little practical values and are not important commercially, because they have serious leaching and stability / reuse problems.
  • Figures 4 and 10 in this manuscript might have copyright problems because they are reproduced without the permission of copyright owners.
  • In section 2, too many paragraphs are spent to review the applications of polyglycerols, which are not the focus points of this Journal.
  • In section 5, too many statements are spent on some well known and predictable results. For example, it is well known and predictable that conversion increases and di- or tri- glycerol selectivities decrease with increasing reaction temperature and time.
  • The number for Conclusion section (page 17) is incorrect. It should be 6.

Author Response

The authors are very grateful to the editor and the reviewers for their comments that helped to significantly improve the manuscript. We have addressed all the comments raised and have modified the paper accordingly. Changes have been highlighted in yellow in the revised manuscript. Below are point-by point responses to the reviewers’ comments and recommendations.

Response to reviewers’ comments

Reviewer #1:

  1. The subject has been reviewed this year by Chong et al. (please see reference 10: A review over the role of catalysts for selective short-chain polyglycerols and other value-added products).

The mentioned review by Chong et al. (2020) discussed shape selective catalysts, i.e, for obtaining selectively di- and triglycerol, mainly focusing on layered clay catalysts; whereas, in our review, we cover the whole range of catalysts, not only to obtain short chain polyglycerols but also higher degree of PGs. In additions, we address the critical issue of leaching and the stability of such heterogenous catalysts, which was missing in the previous reviews. Finally, the effect of reaction conditions has also been described in our review.

  1. The heterogeneous alkaline reviewed in this manuscript have little practical values and are not important commercially, because they have serious leaching and stability / reuse problems.

First of all, as also mentioned in section 3.3, so far, all heterogenous catalysts used in glycerol polymerization reaction have stability issues due to the reaction conditions. In the present review, we have tried to highlight this issue which has been neglected in the literature so far. We are convinced that this is very important in order to give a brighter view for the researcher in this filed to develop new alkaline heterogenous catalysts that could overcome this issue.

Secondly, alkaline-based heterogeneous catalysts, especially the calcium-based ones, could be very attractive candidates for the industry due to their wide availability, their low cost and their absence of toxicity provided leaching issues are cleared.

  1. Figures 4 and 10 in this manuscript might have copyright problems because they are reproduced without the permission of copyright owners.

Referring to the copyright instruction for MDPI (https://www.mdpi.com/authors/rights):  « Graphs, Charts, Schemes and Artworks that are completely redrawn by the authors and significantly changed beyond recognition do not require permission », we only required the copyright permission for figure 9.

  1. In section 2, too many paragraphs are spent to review the applications of polyglycerols, which are not the focus points of this Journal.

We have briefly explained the application of polyglycerols in the introduction, to express the importance of these polymers in the various industries, besides, to highlight the demanded criteria in each sector in the terms of polymer length, etc.

  1. In section 5, too many statements are spent on some well known and predictable results. For example, it is well known and predictable that conversion increases and di- or tri- glycerol selectivities decrease with increasing reaction temperature and time.

We thank the reviewer for this very good point. In general speaking, conversion and the degree of polymerization would increase by rising the temperature and time. However, we believe that it is still important to describe some examples of the effect of these parameters on polymerization, as this review could be of interest for researchers who would like to know about actual trends and behaviour of the catalysts used so far.

  1. The number for Conclusion section (page 17) is incorrect. It should be 6.

We thank the reviewer for the careful reading and we have accordingly corrected the text.

Reviewer 2 Report

Manuscript ID : catalysts-895324

Manuscript Title: Alkaline-based catalysts for glycerol polymerization reaction: a review

This is a great and nicely written review making scope of different aspects of the synthesis, most crucial properties and applications of poly(glycerol) polymers PGs. This review will be of interest to the communities of polymer chemists and of chemists developing green chemical approaches. Also, the instructive content of this contribution should be very helpful and informative for undergraduates and freshmen having interest in studying applications of ecofriendly technologies for the production of polymer-based chemical and materials for bio-sources.
 A minor drawback, which has been overlooked or has not been highlighted specifically, is the lack of a critical discussion on the point if the average polymerization degree (Xn or DPn) of the PGs synthesized by polycondensation (step-growth mechanism) could be controlled through the relationship with the conversion of polymer (extent of reaction p) as described by the equation of Carothers (Xn = 1/(1 -p)), and/or by other factors.

Author Response

The authors are very grateful to the editor and the reviewers for their comments that helped to significantly improve the manuscript. We have addressed all the comments raised and have modified the paper accordingly. Changes have been highlighted in yellow in the revised manuscript. Below are point-by point responses to the reviewers’ comments and recommendations.

Reviewer 2:

  1. A minor drawback, which has been overlooked or has not been highlighted specifically, is the lack of a critical discussion on the point if the average polymerization degree (Xn or DPn) of the PGs synthesized by polycondensation (step-growth mechanism) could be controlled through the relationship with the conversion of polymer (extent of reaction p) as described by the equation of Carothers (Xn = 1/(1 -p)), and/or by other factors.

We fully agree with the reviewer. In fact, the glycerol self-condensation reactions, as also mentioned in this review, are influenced by many factors and various possible mechanisms depending on the catalyst types, etc. As a consequence, the average degree of polymerization could be affected directly by several factors including conversion. Hence, in this case, using an equation such as Carothers equation would not describe the real average polymerization degree. Discussion has been added to the text.

Reviewer 3 Report

The manuscript describes the review on the synthesis of polygrycerols including the reaction conditions and the reaction mechanisms. Some of the important applications of polygrycerols are also mentioned. The following points should be considered.

  1. In figure 2, Solvay PG3 data were used as a reference. Structure of Solvay-PG3 should be specified or explained clearly in the text.
  2. Explanation about the horizontal axis of Figure 2 should also be required. For example, if PG5 means pentaglycerol, I want to know how to purify and isolate the pure PG5.
  3. Control of the structure of polyglycerols should be mentioned. For example, how linear polyglycerls are obtained.
  4. In Figure 3, various kinds of polyglycerol structure were shown. The authors should describe how these structures were detected and confirmed.
  5. For Figure 4, the use of colored dots may be better to recognize each catalyst performance.
  6. For the heterogeneous catalyst system summarized in Table 3, time-conversion curves like Figure 3 may be easy to understand the catalytic activity.

Author Response

Please, see our comments in the attached file (text + figure)

Round 2

Reviewer 1 Report

The revised manuscript has been improved.

Author Response

Thank you very much!

Reviewer 3 Report

There are still structure ambiguity in the manuscript.

For example, in figure 1, the products must contain their isomers. The authors ignore them.

If the products obtained were only the specified ones, the purification processes or isolation process must be explained. Many readers may be very much interested in the process.

In Figure 2, we still do not know what Solvay-PG3 is. Relationship between Polyglycerol-3 and Solvey-PG3 is not clarified in the text. Moreover, Figure 2 legend includes another word, Solvay PG3 (without hyphen). Structure of PG2, PG3, ……PG10 are not specified. If these are the same as those in Table 1, the authors should mention clearly in Figure 2.

It is impossible to discuss chemistry without molecular structure. I think the authors should clarify the structure variety of oligo glycerol compounds before the discussion.

Author Response

Thank you very much for the time you spent for re-checking our manuscript.

Here are our answers:

  1. In figure 1, the products must contain their isomers. The authors ignore them. If the products obtained were only the specified ones, the purification processes or isolation process must be explained. Many readers may be very much interested in the process.

We fully agree with the reviewer. In fact, the products shown in Figure 1 (now in Figure 3) are the desired products (linear forms). Hence, cyclic and hybrid cyclic-linear forms were added to this figure to illustrate the diversity of possible products.

It should be also noted that we draw a general form of polyglycerol with “n” chains, to express the polymerization term, due to the fact that there are numerous PGn isomers (e.g., 12 isomeric compounds for n = 3, 360 for n = 5). It is of course impossible to present all of them all. Regarding the “purification process”, it should be noted that, in the literature, only the separation of undesired compounds that cause colour, odour or a taste to the polymers is discussed. In other words, the purification step has been applied to have a colourless and odourless mixture as mentioned in sections 2 and 3.

  1. In Figure 2, we still do not know what Solvay-PG3 is. Relationship between Polyglycerol-3 and Solvey-PG3 is not clarified in the text.

Polyglycerol-3 means PG3 or triglycerol. To avoid any confusion for the readers, it has been changed to “PG3” in the text.

Secondly, Solvay-PG3 is a commercial name for PG3 manufactured by Solvay Co. But it should be noted that Solvay-PG3 is not pure PG3. As mentioned in the text: “PG3 manufactured by Solvay contains a minimum of 80% di-, tri- and tetraglycerol”.

  1. Moreover, Figure 2 legend includes another word, Solvay PG3 (without hyphen).

We thank the reviewer for the careful reading and we have accordingly corrected the text.

  1. Structure of PG2, PG3, ……PG10 are not specified. If these are the same as those in Table 1, the authors should mention clearly in Figure 2.

It should be noted that the terms “polyglycerol-n” or “PGn” are the general terms for a polyglycerol with “n” units of glycerol, including linear, branched and cyclic. Moreover, as previously mentioned, there is a great number of PGn isomers (19,958,400 isomers for PG10), therefore, once again, it is impossible to determine the precise structure of all PGx in presence. Hence, many authors for simplicity applied the general term PGx for the all possible obtained isomers of polyglycerols with a polymerization degree of X.

In regards to Table 1, as also mentioned in the text that the presented data are only reported for linear PGs.

  1. It is impossible to discuss chemistry without molecular structure. I think the authors should clarify the structure variety of oligo glycerol compounds before the discussion.

The different possible structures have been explained in Section 2, prior to the discussion.

Round 3

Reviewer 3 Report

If the curly arrows show the movement of electron pairs, it is difficult to understand the movement of electron pair from glycerol to OH- in Figure 4.

Figure 3 is not visible in the revised manuscript. If the Figure 6 in the original submission is Figure 3 in the revised version, the curly arrows should be carefully checked.

Author Response

We thank reviewer#3 for careful rechecking of our manuscript.

  1. If the curly arrows show the movement of electron pairs, it is difficult to understand the movement of electron pair from glycerol to OH- in Figure 4.

We suppose that the reviewer means Figure 5 (Reaction mechanism for glycerol polymerization over earth metal oxides catalysts); in this case, Figure 5 has been actually modified to make it clearer.

  1. Figure 3 is not visible in the revised manuscript.

We thank the reviewer for the careful attention and we have accordingly re-inserted the figure with a better resolution.

  1. If the Figure 6 in the original submission is Figure 3 in the revised version, the curly arrows should be carefully checked.

In the revised version (2nd round), only Figure 3 was changed to Figure 1; Figure 1 to Figure 2; Figure 2 to Figure 3. Other figures are still in previous numbering as the original submission.

The curly arrows in Figure 6 have been adjusted.